# The Current Status, Challenges, and Future Potential of Therapeutic Vaccination in Glioblastoma

**DOI:** 10.3390/pharmaceutics15041134

**Published:** 2023-04-03

**Authors:** Bryan J. Neth, Mason J. Webb, Ian F. Parney, Ugur T. Sener

**Affiliations:** 1Department of Neurology, Mayo Clinic, Rochester, MN 55905, USA; 2Department of Medical Oncology, Mayo Clinic, Rochester, MN 55905, USA; 3Department of Neurosurgery, Mayo Clinic, Rochester, MN 55905, USA

**Keywords:** glioblastoma, GBM, vaccine, dendritic cell, adjuvant

## Abstract

Glioblastoma (GBM) is the most common malignant primary brain tumor and confers a dismal prognosis. With only two FDA-approved therapeutics showing modest survival gains since 2005, there is a great need for the development of other disease-targeted therapies. Due, in part, to the profound immunosuppressive microenvironment seen in GBMs, there has been a broad interest in immunotherapy. In both GBMs and other cancers, therapeutic vaccines have generally yielded limited efficacy, despite their theoretical basis. However, recent results from the DCVax-L trial provide some promise for vaccine therapy in GBMs. There is also the potential that future combination therapies with vaccines and adjuvant immunomodulating agents may greatly enhance antitumor immune responses. Clinicians must remain open to novel therapeutic strategies, such as vaccinations, and carefully await the results of ongoing and future trials. In this review of GBM management, the promise and challenges of immunotherapy with a focus on therapeutic vaccinations are discussed. Additionally, adjuvant therapies, logistical considerations, and future directions are discussed.

## 1. Background

Glioblastoma (GBM) is the most common malignant primary brain tumor in adults, representing 14.2% of all central nervous system (CNS) tumors and 50.1% of all malignant tumors [1]. GBM is associated with a high symptom burden and a wide range of neurological symptoms, including cognitive deficits, focal weakness, headaches, and seizures, depending on the tumor’s location [2]. The median overall survival (mOS) is poor at 15 months, despite maximal standard of care therapies [3,4,5]. Only 6.9% of patients survive five years post-diagnosis [1]. This has been unchanged at the population level since 2011 [6]. Despite extensive research, since 2005, only the following two U.S. Food and Drug Administration (FDA)-approved therapeutics have been shown to confer a survival benefit for GBM [7]: the oral chemotherapy agent temozolomide [8] and the tumor-treating fields (TTF) device Optune [9].

The standard of care management for newly diagnosed GBMs begins with a maximal safe surgical resection, followed by radiation therapy with concurrent and adjuvant temozolomide, with or without TTF [7,10,11]. The treatment is tailored according to the patient’s age, performance status, and tumor molecular profile [7]. The standard course of radiation treatment is administered over six weeks [10]. Temozolomide is typically administered concurrently with radiation and in six adjuvant cycles [10]. For patients with a good performance status and O6-methylguanine-DNA methyltransferase (*MGMT*)-methylated tumors, the addition of lomustine to their treatment with radiation and temozolomide may confer additional benefits [12]. In elderly patients or patients with a poor functional status, abbreviated courses of radiation therapy can be considered [13]. In similar patients with *MGMT*-methylated tumors, temozolomide monotherapy is an accepted option [14]. The approach to treatment for newly diagnosed GBMs is summarized in Figure 1. Since none of these treatments is curative, clinical trial enrollment should be considered for all patients [7]. Unfortunately, recurrence is almost inevitable for all patients [1]. The median time to the first recurrence after the diagnosis of GBM is 7 months [10]. To date, there is no proven therapy for improving survival in this setting, despite hundreds of clinical trials since the early 2000s [6,7,15]. The therapeutic options for recurrent diseases include the following: clinical trials, further alkylating chemotherapy (i.e., lomustine/CCNU or temozolomide rechallenge), bevacizumab, and regorafenib, with some selected cases of off-label use with immune checkpoint inhibitors, small-molecule-targeted therapy (i.e., EGFR inhibitors), or re-irradiation, with variable practice patterns across institutions [7].

The reasons for this poor therapeutic response are multifaceted [16]. GBM has well-documented intertumoral and intratumoral heterogeneity that serves as a foundation for resistance to therapy [17,18,19,20]. Glioblastoma was the first systematically studied cancer type as part of The Cancer Genome Atlas Research Network (TCGA), initially leading to the identification of proneural, neural, mesenchymal, and classical molecular tumor subtypes [21,22]. Further studies led to the characterization of three subtypes (proneural, mesenchymal, and classical), with a proneural to mesenchymal transition also described for recurrent tumors [23]. Proneural tumors were initially characterized by platelet-derived growth factor receptor alpha (PDGFRA) and isocitrate dehydrogenase (IDH) mutations [24]. IDH mutant tumors were later grouped separately under a more recent World Health Organization (WHO) tumor classification [24,25]. The mesenchymal and classical subtypes were characterized by neurofibromatosis 1 (NF1) mutations and epidermal growth factor receptor (EGFR) amplifications, respectively [24]. The TCGA studies provided tremendous insight into the intertumoral GBM heterogeneity, demonstrating the presence of different driver mutations for different tumors and raising the possibility of rational molecularly based and highly individualized treatments [23]. Unfortunately, to date, subtype classification has not translated into effective treatment stratifications or improved outcomes, in part due to intratumoral heterogeneity and phenotype switching [26,27]. Traditional chemotherapy and targeted molecular therapeutic options may only work on a subpopulation of the tumor and, ultimately, negatively select for resistant clonal subpopulations [26] or cause phenotype switching, a phenomenon whereby tumor cells transition their invasiveness and/or differentiation [27]. The normal brain architecture itself also serves as a challenge. Even if a promising systemic therapy option is identified, its ability to cross the blood–brain barrier [28] and penetrate within the brain and tumor itself remains a challenge [29,30], such that there has been recent focus on novel strategies (i.e., ultrasound) to open the blood–brain barrier for drug delivery [31]. Among the most important factors in GBM’s resistance to treatment in an immunotherapeutic era is a profound local and systemic immunosuppressive state [32,33,34].

## 2. Immunosuppressive State and Immune-Targeted Therapies

Several mechanisms contribute to GBM immune evasion, including increased PD-L1 expression [35], release of cytokines (IL-10, IL-6, and TGF-beta1), and, potentially, downregulation of MHC expression [36,37]. This is compounded by aspects related to the tumor microenvironment, including tumor-associated microglia/macrophages, secretion of protumorigenic/survival factors (IL-10 and IDO), vascular-related factors (VEGF, FGF, and pericyte proliferation), and release of immunosuppressive extracellular vesicles [36,38,39]. The GBM microenvironment is increasingly recognized as an important factor in the development of promising therapeutics for GBM. Out of the scope of the present review, several excellent reviews discuss the implications of the GBM microenvironment on the lack of therapeutic success, especially with immunotherapy [40,41,42]. Despite the fact that GBM tumors rarely metastasize outside the central nervous system, patients with GBM exhibit profound systemic immunosuppression, characterized by decreased overall T-cell numbers and function [43] combined with increased circulating immunosuppressive leukocytes, including regulatory T cells and myeloid-derived suppressor cells [44,45,46]. Given the role of immune dysfunction in GBM and the paucity of therapies improving longevity, there has been great interest in developing immune-targeted therapies for GBM [36,38]. Potential therapies fall into the following key classes: (1) immune checkpoint inhibitors (i.e., pembrolizumab, nivolumab, and ipilimumab), (2) T-cell-targeted therapy (i.e., CAR-T), (3) oncolytic viral therapies (i.e., adenovirus leading to cell lysis/death, specifically targeting GBM cell populations), (4) therapeutic vaccines, and (5) cytokine-based therapies [38,47,48,49,50]. These approaches are thoroughly reviewed in a recent manuscript from our group [47]. The focus of the present manuscript will be on therapeutic vaccines for GBM.

There has been great interest in harnessing active tumor-specific immune responses as a disease-targeting therapy for cancers [51], in addition to the non-specific reduction in tumor-mediated immunosuppression. For example, therapeutic cancer vaccines expose antigen-presenting cells to cancer-specific antigens, thereby inducing a cytotoxic T-cell response and ultimately cancer cell death [52,53,54]. However, there are only a few FDA-approved cancer therapeutic vaccines, despite extensive preclinical development and numerous clinical trials [53]. Bacillus Calmette-Guérin (BCG) is approved for patients with early-stage bladder cancer [55,56]. Sipuleucel-T is a dendritic cell (DC)-based vaccine approved in 2010 for use in men with metastatic castration-resistant prostate cancer [57]. Talimogene laherparepvec T-VEC (Imlygic^®^) was approved for the treatment of metastatic melanomas [58]. Given the promise of immune therapy in targeting a broad spectrum of malignancies, there has been great interest in the development of a therapeutic vaccine for GBM [59,60].

## 3. Therapeutic Vaccines

Therapeutic cancer vaccines may be categorized based on the target antigen and vehicle or mechanism of delivery. Peptide vaccines are by far the most studied antigen platform in GBM [52,53,61]. Peptides consist of amino acid chains, which are ultimately presented to T cells in lymphoid tissues by DC, priming an immune response. The peptide antigen targets vary in size and number (single or multipeptide) [53,62]. There is also interest in nucleic acid vaccines based on bacterial DNA plasmids and mRNA [63,64]. Once delivered, DNA plasmids enter the nucleus, whereby the target antigen is transcribed and expressed on the target cell through MHC class I/II presentation, generating innate and adaptive immune responses [63]. mRNA vaccines consist of single-stranded mRNA transcripts that encode antigen(s) of interest, which are incorporated into antigen-presenting cells (APCs) that are subsequently activated to generate an immune response, similar to DNA plasmid vaccines [64].

GBM vaccine delivery is often accomplished through DCs [61]. DCs are APCs with an important role in normal immune function [65]. For vaccine development, DCs are exposed to an antigen of interest ex vivo and matured prior to injection. The DCs then migrate to lymphoid tissues, leading to the activation of T cells and other immune mediators. This prompts the immune system to target the antigen of interest, in this case, one common to the primary tumor [66]. Heat-shock proteins (HSPs) [67] are less commonly used vaccine vehicles in GBMs. HSPs are produced in response to various stressors to mitigate the downstream effects of misfolded protein production by assisting protein refolding. If the functional proteins cannot be refolded, HSPs chaperone these misfolded proteins to proteasomes for destruction [67,68]. In vaccine development, HSPs are combined with the peptide of interest, which leads to a T-cell response [69].

In this review, a selection of the most promising therapeutic vaccines in GBM are discussed, with a complete list of ongoing and future clinical trials shown in Table 1 (as of January 2023; listed on clinicaltrials.gov). The annual distribution of GBM vaccine trials is presented in Figure 2.

Multiple DC vaccines have been developed for use in GBMs [70,71,72]. Of these, DCVax-L utilizes an autologous tumor lysate, developed by Northwest Biotherapeutics, Inc., which has been under development as part of a larger DCVax platform. The results from a Phase 3 trial (NCT00045968) have recently been published, which assessed the impact of DCVax-L on the survival rate in patients with newly diagnosed and recurrent GBM who otherwise received standard of care [73] and updated interim analyses [74]. The nonrandomized, externally controlled trial took place from August 2007 to November 2015 at 94 sites in four countries (the USA, Canada, the UK, and Germany). A total of 331 patients with newly diagnosed GBM were enrolled (with a median age of 56 years; 61% were male; 89% were white). They had an mOS (*n* = 232) of 19.3 months (95% CI, 17.5–21.3) from the time of randomization (22.4 months from surgery) in the DCVax-L cohort relative to 16.5 months (95% CI, 16.0–17.5) from randomization in external controls, with a hazard ratio (HR) of 0.8 (98% CI, 0–0.94, *p* = 0.002). The mOS in patients with recurrent GBM (*n* = 64) was 13.2 months (95% CI, 9.7–16.8) from relapse versus 7.8 months (95% CI, 7.2–8.2) in the external controls, with an HR of 0.58 (98% CI, 0–0.76, *p* < 0.001). In addition to this observed survival benefit in both the newly diagnosed and recurrent GBMs, the authors noted a greater proportion of long-term survivors (36–60 months) in the DCVax-L group. Predefined subgroup analyses showed a potential survival benefit, particularly in patients that were 65 years of age or older, those with subtotal resection at the time of surgery, and those with MGMT promoter methylation. Importantly, DCVax-L showed negligible toxicity, with only five serious adverse events out of 2151 total administered doses. Although the results of this Phase 3 trial are promising, they must be interpreted in light of several limitations. External controls without individual patient-level data were used as a comparison for efficacy. The primary endpoint was changed from the initial design due to prominent radiographic changes after therapy (i.e., radiographic or pseudoprogression). Temozolomide, which may dampen the immune response, was given to most patients [75,76]. Lastly, the DCVax-L used for the recurrent GBM was derived from a tumor at initial diagnosis. While there are likely to be some similarities in the tumor at recurrence, recurrent GBM evolves with changes in the most prominent clonal subpopulations and phenotype switching [26,27].

Survivin (or BIRC5) is an anti-apoptotic protein that inhibits caspase activation and is highly expressed in most cancers, including GBM [77]. BIRC5 expression in GBM is associated with a worse prognosis; however, it is not present in normal glial tissues, making it an ideal vaccine target [78,79]. SurVaxM is a synthetic survivin vaccine, recently studied in early-phase clinical trials. An early clinical study of 9 patients (NCT01250470) demonstrated safety (mostly Grade 1 and no serious adverse events) [80], and the results from a recent Phase 2a trial (NCT02455557) of 64 patients with newly diagnosed GBM confirmed the previous safety data [81]. The survival data demonstrated a median progression-free survival (PFS) of 11.4 months (95.2% of the patients remained progression-free after 6 months), and the mOS was 25.9 months from the time of the first SurVaxM dose [81]. These results have since led to a larger ongoing Phase 2 trial (SURVIVE; NCT05163080), with an estimated completion date of April 2024.

The epidermal growth factor receptor (EGFR) is amplified in approximately 40% of GBMs, with an estimated 20% of GBMs harboring the mutant EGFRvIII, leading to the activation of the signaling pathways and contributing to malignant potential [82,83]. A Phase 3, randomized, double-blind, placebo-controlled clinical trial (Act IV; NCT01480479) of rindopepimut, an EGFRvIII-targeting peptide vaccine, in patients with newly diagnosed GBM (total *n* = 745; vaccine *n* = 371; placebo *n* = 374) was terminated for futility after a preplanned interim analysis [84]. There was no improvement in the mOS between the rindopepimut (20.1 months; 95% CI 18.5–22.1) versus the placebo (20.0 months; 95% CI 18.1–21.9), with an HR of 1.01 (95% CI 0.79–1.30, *p* = 0.93) [84]. The loss of EGFRvIII expression was described in about 57–59% of the tumors in both the treatment and control arms of the study [85], highlighting the frequency of phenotype switching and its importance when developing molecularly targeted therapeutics for GBMs.

IMA950 is a multipeptide vaccine that contains the following nine MHC class I-restricted peptides: brevican (BCAN), chondroitin sulfate proteoglycan 4 (CSPG4), fatty-acid-binding protein 7 (FABP7), insulin-like growth factor 2 mRNA-binding protein 3 (IGF2BP3), neuronal cell adhesion molecules (NRCAMs), neuroligin 4 X-linked (NLGN4X), protein tyrosine phosphatase, receptor type Z1 (PTPRZ1), and tenascin C (TNC). These antigens are typically overexpressed on the surface of GBM tumor samples and absent in normal glial tissues [86]. It is important to note that with a multipeptide approach, assessing the response may be more challenging than with more targeted peptide approaches. Immunogenicity of individual peptides in the form of a sustained T-cell adaptive response, as well as autocrine effects mediated by the secretion of cytokines and lymphokines by antigen-presenting cells, requires further preclinical and clinical characterization. IMA950 also contains two MHC class II-restricted peptides, c-Met and survivin, which are overexpressed in GBM but not expressed on the cell surface [87]. A recent Phase 1/2 clinical trial assessed IMA950 in combination with adjuvant poly-ICLC in 16 patients with newly diagnosed GBM. This trial (NCT01920191) demonstrated safety (although four patients had short-term cerebral edema with quick recovery) and immunogenicity, with an mOS of 19 months [87], supporting the results from a previous trial (NCT01222221) [88]. An ongoing Phase 1/2 trial is assessing IMA950 and poly-ICLC in combination with the immune checkpoint inhibitor (ICI) pembrolizumab (NCT03665545).

Cytomegalovirus (CMV) proteins have been found in the majority of GBMs [89], with the CMV phosphoprotein 65 (pp65) commonly expressed in tumors but not in normal glial tissues [90]. There has been great interest in CMV pp65-based vaccines, with some promising results in early-phase trials. A single-arm Phase 1 trial assessing a CMV pp65 vaccine in combination with dose-dense temozolomide in 11 patients with newly diagnosed GBM showed a greatly improved PFS (mPFS of 25.3 months; 95% CI 11–∞) and OS (mOS of 41.1 months; 95% CI 21.6–∞) from historical controls. There were also four long-term survivors (36%), who remained progression-free at 59+ months (NCT00639639) [91]. Given these impressive results, several trials assessing CMV pp65-based vaccines alone or in combination with therapy are ongoing and/or soon to be reported as of January 2023 (NCT02465268, NCT04963413, NCT04741984, NCT04573140, NCT00639639, NCT05283109, NCT03382977, and NCT03299309).

Wilm’s tumor 1 (WT1) is a transcription factor found in various malignancies, including GBM [92]. A Phase 1 dose-escalation trial of DSP-7888 (NCT02498665), a peptide vaccine including two synthetic peptides derived from WT1, was completed in patients with multiple malignancies (pancreatic cancers, sarcomas, non-small-cell lung cancers, ovarian cancers, and melanomas), including GBM (*n* = 7). The results from this trial demonstrated safety (the most common adverse events were low-grade injection site reactions without dose-limiting toxicity) [93], supporting further study in an ongoing Phase 3 trial (NCT03149003) in patients with recurrent or progressive GBMs.

EO2401 is an off-the-shelf, microbiome-derived, multipeptide vaccine that combines peptides that mimic cancer-driver antigens (IL13Ra2, BIRC5, and FOXM1) and the helper peptide UCP2, which is currently being assessed in a Phase 1B/2A trial of patients with progressive GBM +/− ICI/bevacizumab therapy (ROSALIE, NCT04116658) [94,95]. The interim results suggest that the treatment leads to strong immunogenicity with an mPFS of 1.8 months and a 6-month OS of 85% in 40 patients [95], with a further improved PFS (5.5 months) and tumor response with the addition of bevacizumab to EO2401 + ICI (nivolumab) [94].

VXM01 is a DNA plasmid vaccine that contains an attenuated strain of *Salmonella typhimurium*, which encodes the murine vascular endothelial growth factor receptor 2 (VEGFR-2). VEGFR-2 activation upregulates angiogenesis and cell proliferation, both necessary for tumor growth, and is commonly expressed within tumor microenvironments. Immunologically targeting and inhibiting this receptor likewise impairs tumor progression [96]. A clinical study (*n* = 14; NCT02718443) of VXM01 in progressive GBMs showed a favorable response in five patients, and the prolonged survivors had lower intratumoral PD-L1 expression, suggesting that combination with ICI therapy may boost the response to VXM01 therapy [97]. An ongoing Phase 1/2 trial (NCT03750071) is assessing VXM01 in combination with the ICI avelumab (a PD-L1 inhibitor) in recurrent GBMs.

There is increasing interest in personalized approaches to vaccination [98]. A Phase 1 trial (GAPVAC-101; NCT02149225) demonstrated overall safety (three serious adverse events attributable to the study drug—one cerebral edema and two anaphylaxis) and immunogenicity (sustained responses of CD8+ and CD4+ T cells) in 15 patients with newly diagnosed GBM [99]. These patients were administered APVAC1, derived from premanufactured unmutated tumor antigens, followed by APVAC2, derived from targeted tumor neoepitopes personalized from mutations in individual tumors (APVAC1: CD8+; APVAC2: CD4+). This trial resulted in an mPFS of 14.2 months and an mOS of 29.0 months [99]. Another ongoing Phase 1 trial is assessing a personalized vaccine (NeoVax) with an ICI (pembrolizumab) in GBM (NCT02287428).

## 4. Delivery

The method of vaccine delivery is also a consideration in the development of cancer therapeutic vaccines [100,101,102]. The vast majority of vaccination trials in glioblastomas utilize subcutaneous, intradermal, or intravenous delivery methods (Table 1). This is reasonable in that antigen exposure leads to a systemic immune response that ultimately targets the tumor [66]. However, there is increasing interest in novel drug delivery methods that may improve efficacy. Some examples of local delivery include the following: polymeric wafers, nanofibrous scaffolds, and hydrogels [100,101,102]. While out of the scope of the present review, recent articles extensively review advances in drug delivery that may be incorporated into vaccine development for GBM [100,101,102].

## 5. Challenges

Several factors likely contribute to the failure of most therapeutic vaccines to impact clinically meaningful endpoints [36,52,53]. A candidate vaccine may not lead to robust and sustained immunogenicity, which may be due to the antigen selection or the tumor microenvironment. There is an increasing understanding that GBM is “immunologically cold”, meaning that it is poorly infiltrated by effector cell populations and contains highly suppressive microenvironments. This, in turn, may cause GBM to be less responsive to immunotherapy, particularly monotherapy with ICIs [53,61,103]. GBM leads to a systemic immunosuppressive state [32,34], which should be considered in the design of immunotherapy. This may be further compounded by immunosuppressive therapies, such as temozolomide, which reduce the potential beneficial effects of vaccines. Indeed, alkylating chemotherapy leads to a meaningful and lasting impairment of T- and B-cell responses and proliferative potentials [75,76]. Adjuvant immunotherapies have been increasingly utilized to potentiate a more robust immune response [104,105,106]. Another important factor contributing to the challenges of immunotherapy and glioblastomas is iatrogenic immunosuppression with the use of corticosteroids (i.e., dexamethasone). Corticosteroids are often used in clinical practice to improve symptoms secondary to tumor mass effects, edema, and/or treatment effects (i.e., radiation) [107,108]. As the immunosuppressive effect of corticosteroids is well-established, it has been proposed that it has contributed to the failure of specific immunotherapies (immune checkpoint inhibitors) in GBMs [109]. The use of corticosteroids must be considered in the trial design and interpretation of results [110,111].

## 6. Adjuvant Therapies

With the lack of success for nearly all cancer therapeutic vaccines, there has been increasing attention on the use of adjuvant therapies, which may potentiate vaccine efficacy by heightening the immune response to the delivered antigen [53,61,104,105]. Multiple classes of adjuvant therapy are increasingly combined. For a more thorough review of adjuvant therapies utilized in peptide-based vaccines, see several excellent references in [104,105,106]. The key adjuvants studied in GBM trials are discussed below.

Bevacizumab is a monoclonal antibody used against the vascular endothelial growth factor (VEGF) and has been FDA-approved for use in GBM [112]. Its primary mechanism is as an antiangiogenic and reducer of cerebral edema. Bevacizumab may also reduce immunosuppression by improving VEGF-associated dysfunction in antigen presentation, lymphocyte trafficking, and DC maturation [36,113,114].

Montanide is a water-in-oil emulsion that works by prolonging the release of vaccine antigens. Two primary formulations are used in humans, primarily Montanide ISA-51 and Montanide ISA-720, which differ based on the included oil [115]. A Phase 2 trial in 64 patients with newly diagnosed GBM assessed SurVaxM in Montanide ISA-51 (plus sargramostim and concurrent/adjuvant temozolomide) and showed no serious adverse events related to the vaccine/Montanide (NCT02455557) [81]. Montanide ISA-51 was also used as adjunctive immunotherapy in an early-phase trial (NOA16; *n* = 32; NCT02454634) assessing the safety of an IDH1(R132H)-specific peptide vaccine in patients with another primary central nervous system tumor (IDH1 mutant glioma). This trial demonstrated safety (one serious adverse event; no regime-limiting toxicities) and vaccine-induced immune responses in 93% of the participants [116].

Tetanus-diphtheria toxoid (Td) is used as an antigen to precondition vaccination with the goal of promoting a more robust immune response [117,118]. Preclinical and clinical (NCT00639639) data show that a Td treatment prior to a DC vaccine (cytomegalovirus phosphoprotein 65, or CMV pp65) leads to improved DC migration, suppressed tumor growth, and prolonged survival (*n* = 12), which appeared dependent on the increased levels of the chemokine CCL3 [119].

Toll-like receptors (TLRs) are critical for the activation of the innate immune system and can be stimulated by imiquimod, poly-ICLC, and CpG oligonucleotides [106]. Imiquimod is an imidazoquinoline that activates TLR7/8, thereby leading to the production of inflammatory cytokines and enhanced DC migration. Poly-ICLC acts through TLR3, leading to the production of inflammatory cytokines and interferons and the upregulation of costimulatory molecules [106]. CpG oligonucleotides activate TLR9, leading to the release of inflammatory cytokines. There are three classes of CpG oligonucleotides (CpG A, B, and C) with varying properties.

While inflammatory cytokines are produced through TLR activation, various cytokines may also be administered directly as an adjunctive approach to stimulate the innate and/or adaptive immune system [120]. Interferon alpha has been approved for melanomas, and high-dose interleukin-2 (HDIL-2) has been approved for renal cell carcinomas and melanomas [121,122]. The granulocyte-macrophage colony-stimulating factor (GM-CSF) leads to DC recruitment and maturation with the activation of other immune cells (NK cells, macrophages, and neutrophils) [106,123]. While cytokines can promote immune activation, this class of therapies is most often used as an adjunct rather than a single agent [120]. Of the cytokines, GM-CSF and IL-12 are currently the most studied in GBM.

ICIs are FDA-approved [124] for use in many solid tumors. The primary mechanism of reversing the T-cell inhibitory effects of either the cytotoxic T-lymphocyte-associated protein 4 (CTLA-4) or the programmed cell death protein 1/programmed cell death ligand 1 (PD-1/PD-L1) has been extensively reviewed [124,125,126,127]. While they have been ineffective in newly diagnosed/recurrent GBM trials to date [47], they may be used to augment the immune response when combined with other immune-modulating agents [128,129].

Indoleamine 2,3-dioxygenase 1 (IDO1) has been shown to promote cancer immune escape by catalyzing the breakdown of tryptophan to kynurenine. This is the initial step of the kynurenine pathway that has immunoregulatory functions [130,131]. High expressions of IDO1 have been associated with a worse prognosis in several malignancies [130], including GBM [132,133]. The use of a novel IDO1 inhibitor, BGB-5777, showed survival benefits in a mouse GBM model when combined with ICI and RT [134], forming the basis for ongoing/future trials (combined with ICI and RT +/− other therapies), including BMS-986205 (NCT04047706) and epacadostat (NCT03532295).

## 7. Logistical Considerations

It is important to consider the inevitable logistical challenges that arise from the more widespread use of vaccine therapies. DCVax-L, for example, is a personalized vaccine that uses an autologous tumor homogenate and, therefore, represents a highly individualized therapy [135,136]. Several steps are needed for broad implementation. Each institution performing resection would need to provide enough tissue to the vaccine manufacturer for antigen loading and the ultimate maturation of the DC for delivery. Patients would need to undergo leukapheresis prior to the start of standard therapy, potentially delaying treatment initiation. Multiple resections may be needed to ensure that the individualized vaccine is designed against antigens relevant to the recurrent rather than the primary tumor [26,27]. The feasibility of such an approach is also likely to hamper widespread adoption unless there is a robust survival advantage or improvement in the quality of life/functional status. The logistical challenges with an autologous antigen DC vaccine are lessened with the use of a standardized vaccine with target antigens commonly found in GBM [135,136,137] or the use of allogeneic tumor lysates [138]. Although there have been limited meaningful clinical data to support these approaches to date [139], several trials are ongoing.

## 8. Conclusions

GBM is the most common malignant primary brain tumor and confers a dismal prognosis. With only two FDA-approved therapeutics showing modest survival gains since 2005, there is a great need for the development of other disease-targeted therapies. Due, in part, to the profound immunosuppressive microenvironment seen in GBM, there has been a broad interest in immunotherapy. In both GBMs and other cancers, therapeutic vaccines have generally yielded limited efficacy, despite their theoretical basis [53]. However, recent results from the DCVax-L trial [73] provide some promise for vaccine therapy in GBMs. There is also potential that future combination therapies with vaccines and adjuvant immunomodulating agents [105,106] may greatly enhance antitumor immune responses [53]. The neuro-oncology community must remain open to novel therapeutic strategies, such as vaccinations, and carefully await the results of ongoing and future trials.

## Figures and Tables

**Figure 1 pharmaceutics-15-01134-f001:**
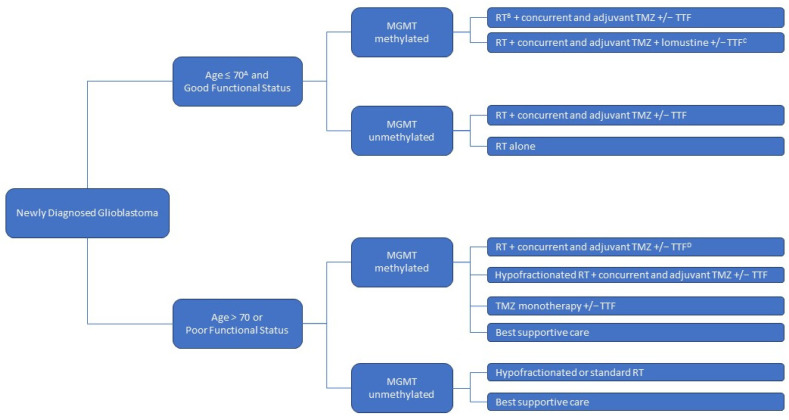
Approach to newly diagnosed glioblastoma treatments, following maximal safe surgical resection. MGMT = O6-methylguanine-DNA methyltransferase; RT = radiation therapy; TMZ = temozolomide; TTF = tumor-treating fields. A: Elderly status is defined as ≥65 in certain studies. B: A standard radiation therapy course is administered over 6 weeks. C: For MGMT-methylated patients with a good performance status, the addition of lomustine to the standard treatment with radiation therapy and temozolomide, with or without tumor-treating fields, may provide additional benefits (see text). D: In selected elderly patients with a good performance status, standard 6-week radiation therapy with concurrent and adjuvant temozolomide can be considered (see text).

**Figure 2 pharmaceutics-15-01134-f002:**
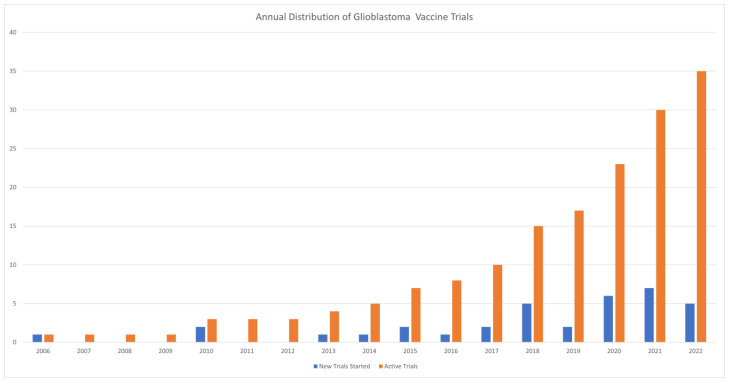
Annual Distribution of Glioblastoma Vaccine Clinical Trials.

**Table 1 pharmaceutics-15-01134-t001:** Active, recruiting, or soon-to-be-recruiting clinical trials investigating therapeutic vaccinations for glioblastomas.

ID	Phase	Design	Patients Enrolled	Diagnosis	Vaccine	Delivery	Adjuvant	Control	Primary Endpoint	Age	Status	Start	Completion
NCT03149003	Phase 3	Rand	236	rGBM	DSP-7888	ID	Bevacizumab	Bevacizumab + SOC	DLT, OS	18+	Active, not recruiting	8 December 2017	1 November 2023
NCT05100641	Phase 3	Rand	726	nGBM	AV-GBM-1	Not listed	GM-CSF	Autologous monocytes	OS	18+	Not yet recruiting	1 March 2022	1 March 2027
NCT04277221	Phase 3	Rand	118	rGBM	ADCTA	SC	Bevacizumab	Bevacizumab + SOC	OS	18–70	Unknown status	19 September 2019	31 December 2022
NCT02455557	Phase 2	Single Arm	66	nGBM	SurVaxM	SC	Montanide ISA 51 and Sargramostim	-	PFS6	18+	Active, not recruiting	4 May 2015	30 December 2023
NCT01204684	Phase 2	Rand	60	n/rGBM	ATL-DC	ID	Resiquimod or Poly-ICLC	-	Most effective combination	18–70	Active, not recruiting	8 October 2010	31 January 2025
NCT03400917	Phase 2	Single Arm	55	nGBM	AV-GBM-1	Not listed	GM-CSF	-	OS	18–70	Active, not recruiting	20 June 2018	1 February 2023
NCT04523688	Phase 2	Single Arm	28	GBM	ATL-DC	ID	-	-	PFS, AEs	18+	Not yet recruiting	1 March 2021	1 December 2025
NCT04888611	Phase 2	Rand	40	rGBM	GSC-DCV	Not listed	Camrelizumab	Camrelizumab alone	OS, PFS	18–70	Recruiting	26 October 2021	1 May 2024
NCT02465268	Phase 2	Rand	175	nGBM	CMV-DC	SC	GM-CSF	Autologous monocytes	OS	18+	Recruiting	1 August 2016	1 June 2024
NCT04280848	Phase 2	Non-Rand	56	nGBM	UCPVax	SC	-	-	Immunogenicity	18–75	Recruiting	26 May 2020	1 May 2023
NCT03395587	Phase 2	Rand	136	nGBM	ATL-DC	ID	-	SOC	OS	18+	Recruiting	6 March 2018	6 June 2025
NCT05163080	Phase 2	Rand	265	nGBM	SurVaxM	SC	Montanide and Sargramostim	Montanide and Sargramostim	OS	18+	Recruiting	18 November 2021	18 April 2024
NCT03382977	Phase 1/2	Non-Rand	98	rGBM	VBI-1901	ID	GM-CSF	SOC	DLT, AEs	18+	Active, not recruiting	6 December 2017	1 August 2025
NCT03665545	Phase 1/2	Rand	18	rGBM	IMA950	SC	Poly-ICLC +/− Pembrolizumab	-	AEs	18+	Active, not recruiting	25 October 2018	31 December 2023
NCT03750071	Phase 1/2	Single Arm	30	rGBM	VXM01	Not listed	Avelumab	-	AEs	18+	Active, not recruiting	21 November 2018	31 December 2022
NCT04116658	Phase 1/2	Non-Rand	52	rGBM	EO2401	Not listed	Nivolumab +/− Bevacizumab	-	AEs	18+	Recruiting	13 July 2020	1 August 2023
NCT02649582	Phase 1/2	Single Arm	20	nGBM	auto-WT1-DC	ID	-	-	OS	18+	Recruiting	1 December 2015	1 December 2024
NCT04801147	Phase 1/2	Single Arm	76	nGBM	ATL-DC	ID	-	-	PFS12	18–70	Recruiting	1 June 2010	1 December 2023
NCT04388033	Phase 1/2	Single Arm	10	nGBM	ATL-DC	ID	IL-12	-	AEs, PFS6	18–75	Recruiting	1 December 2020	1 December 2023
NCT04015700	Phase 1	Single Arm	9	nGBM	GNOS-PV01	Not listed	INO-9012 (IL-12)	-	DLT, Feasibility	18+	Active, not recruiting	14 July 2020	13 April 2023
NCT03223103	Phase 1	Single Arm	13	nGBM	MTA-based Personalized Vaccine	Not listed	Poly-ICLC	-	DLT	18+	Active, not recruiting	1 March 2018	1 May 2023
NCT04642937	Phase 1	Sequential	24	rGBM	GBM6-AD (Allogeneic TL)	Not listed	hP1A8 + Imiquimod	-	MTD	18+	Active, not recruiting	1 December 2020	1 November 2023
NCT00639639	Phase 1	Single Arm	42	nGBM	CMV-DC (auto)	ID	tetanus toxoid	-	Feasibility	18+	Active, not recruiting	1 January 2006	1 December 2022
NCT04741984	Phase 1	Sequential	27	nGBM	MT-201-GBM	IV	-	-	MTD, Immunogenicity	18+	Not yet recruiting	1 October 2022	1 August 2025
NCT05283109	Phase 1	Single Arm	36	nGBM	P30-EPS (P30-linked EphA2, CMV pp65, survivin)	Not listed	Hiltonol	-	DLT	18+	Not yet recruiting	1 November 2022	1 February 2028
NCT04968366	Phase 1	Single Arm	10	nGBM	ATL-DC (with multiple tumor neoantigen peptides)	ID	-	-	AEs	18–75	Recruiting	30 July 2021	1 August 2024
NCT04573140	Phase 1	Single Arm	28	nGBM	RNA-LP	IV	-	-	Feasibility, DLT, MTD	21+	Recruiting	26 October 2021	1 July 2027
NCT04963413	Phase 1	Single Arm	10	nGBM	CMV-DC (auto)	Not listed	GM-CSF	-	Feasibility	18–90	Recruiting	13 January 2022	1 May 2025
NCT02287428	Phase 1	Rand	56	nGBM	NeoVax	Not listed	Pembrolizumab	-	AEs, Feasibility	18+	Recruiting	1 November 2014	1 January 2026
NCT04201873	Phase 1	Rand	40	rGBM	ATL-DC	ID	Pembrolizumab, Poly-ICLC	IV Placebo	Immunogenicity, AEs	18+	Recruiting	8 January 2020	1 August 2025
NCT04552886	Phase 1	Non-Rand	24	nGBM	ATL-DC (Th-1 specific)	Not listed	-	-	AEs	18+	Recruiting	11 October 2021	31 December 2025
NCT04842513	Phase 1	Single Arm	15	nGBM	Multipeptide vaccine	SC	XS15	-	AEs, Immunogenicity	18+	Recruiting	3 May 2021	2 May 2024
NCT05557240	Phase 1	Single Arm	10	nGBM	NeoPep Vaccine1/2 (NPVAC1/2)	Not listed	Poly-ICLC	-	AEs	18–70	Recruiting	13 September 2022	12 August 2025
NCT03360708	Early Phase 1	Single Arm	20	rGBM	ATL-DC + Allo GBM lysate	ID	-	-	AEs, Feasibility	18+	Active, not recruiting	3 June 2019	1 June 2023
NCT01957956	Early Phase 1	Single Arm	21	nGBM	auto-DC + Allo GBM lysate	ID	-	-	AEs	18+	Active, not recruiting	11 November 2013	15 November 2023

Data are taken from clinicaltrials.gov as of January 2023. nGBM = newly diagnosed glioblastoma; rGBM = recurrent glioblastoma; Rand = randomized; Non-Rand = non-randomized; ATL-DC = autologous tumor lysate dendritic cell vaccine; DC = dendritic cell; CMV-DC = CMV-based dendritic cell vaccine; Allo = allogeneic vaccine component; AEs = adverse events; OS = overall survival; PFS = progression-free survival; DLT = dose-limiting toxicity; SOC = standard of care; MTD = maximum tolerated dose; GM-CSF = granulocyte macrophage colony-stimulating factor; ADCTA = autologous dendritic cell/tumor antigen; poly-ICLC = polyinosinic-polycytidylic acid stabilized with polylysine and carboxymethylcellulose; WT1 = Wilms tumor 1; IL-12 = interleukin-12; MTA = mutation-derived tumor antigen; GSC-DC = glioma stem cell dendritic cell; UCPVAx = universal cancer peptide-based vaccination; hP1A8 = human P1A8; P30-EPS = P30-linked Ephrin receptor A2; RNA-LP = ribonucleic acid lipid particle; SC = subcutaneous; ID = intradermal; IV = intravenous.

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
