# Peer review of "The Current Status, Challenges, and Future Potential of Therapeutic Vaccination in Glioblastoma"

_pharmaceutics, 2023, doi:10.3390/pharmaceutics15041134_

Round 1
Reviewer 1 Report
Regarding the manuscript (pharmaceutics- 2255049) entitled:
“The potential of therapeutic vaccination in glioblastoma”
Comments to the Author
General comment
The manuscript covers all the basic principles and clinical trials. Some comment should be considered to improve the manuscript:
1. Many reviews cover similar topics like
https://www.ncbi.nlm.nih.gov/pmc/articles/PMC6163986/
2. More details about the disease and current treatment options should be addressed.
3. Scheme about the disease and therapeutic approached should be added.
4. The delivery systems used should be addressed.
5. Commonly used platforms for localized drug delivery include polymeric wafers, nanofibrous scaffolds, and hydrogels should be added
Author Response
The manuscript covers all the basic principles and clinical trials. Some comment should be considered to improve the manuscript:
- Many reviews cover similar topics like https://www.ncbi.nlm.nih.gov/pmc/articles/PMC6163986/
Authors: Thank you very much for reviewing our manuscript and providing feedback. We included this excellent review as a reference in addition to 35078492 to make our literature review more complete.
- More details about the disease and current treatment options should be addressed.
Authors: We expanded the first paragraph of the Background section to include more recent epidemiology information. Included additional information about the high symptom burden associated with GBM. Included information about median time to recurrence and expanded commentary on challenges of recurrent disease.
- Scheme about the disease and therapeutic approached should be added.
Authors: We added a figure outlining standard of care approach to GBM treatment to address this.
- The delivery systems used should be addressed.
Authors: We have added a column to Table 1 denoting the delivery system for each vaccine.
- Commonly used platforms for localized drug delivery include polymeric wafers, nanofibrous scaffolds, and hydrogels should be added
Authors: We have added an additional section focused on delivery with reference to several other timely reviews that discuss drug delivery, including polymeric wafer, scaffolds, and hydrogels in depth.
Reviewer 2 Report
The authors present a fair and balanced review of the promise and challenges of immunotherapy with focus on therapeutic vaccination for GBM tumours, highlighting very well how treatments for GBM tumours remain scarce representing an area of key unmet need.
It would be recommended to introduce the concept of different GBM subtypes in the background section, when the authors refer to GBM heterogeneity. It is important to acknowledge the existence of different subtypes to explain GBM variability.
In addition to intratumor heterogeneity, it would be important to introduce the concept of inter tumor heterogeneity too. In the logistical considerations section, the authors highlight that having such a personalised treatment approach like DCVax-L poses logical challenges. If on one hand this is true, therapies that are personalised can overcome the inter tumor heterogeneity that is also typical of glioblastoma.
Table 1:
- please make sure that the NCT ID sits on one 1 line only
- several acronyms are not defined in the footnote of the table; suggest adding an explanation/definition for all even if they are defined in the body
- suggest changing enrollment to patients enrolled or number of patients
- suggest changing outcome to endpoint or even better primary endpoint if this is what is reported in the table
Author Response
The authors present a fair and balanced review of the promise and challenges of immunotherapy with focus on therapeutic vaccination for GBM tumours, highlighting very well how treatments for GBM tumours remain scarce representing an area of key unmet need.
Authors: Thank you very much for reviewing our manuscripts and providing feedback. We have addressed the comments below individually.
It would be recommended to introduce the concept of different GBM subtypes in the background section, when the authors refer to GBM heterogeneity. It is important to acknowledge the existence of different subtypes to explain GBM variability.
Authors: We included information about TCGA studies leading to identification of GBM subtypes in the background section to address this.
In addition to intratumor heterogeneity, it would be important to introduce the concept of inter tumor heterogeneity too. In the logistical considerations section, the authors highlight that having such a personalised treatment approach like DCVax-L poses logical challenges. If on one hand this is true, therapies that are personalised can overcome the inter tumor heterogeneity that is also typical of glioblastoma.
Authors: We expanded the background section to introduce the concept of intertumoral heterogeneity and the concept of personalized treatment earlier in the paper. We included information about TCGA studies leading to identification of GBM subtypes, but also acknowledged this has not led to difference in standard of care therapy stratification to date.
Table 1:
- please make sure that the NCT ID sits on one 1 line only
Authors: We updated the table’s formatting to address this.
- several acronyms are not defined in the footnote of the table; suggest adding an explanation/definition for all even if they are defined in the body
Authors: Table footnote has been updated to include the definition of additional acronyms.
- suggest changing enrollment to patients enrolled or number of patients
Authors: Changed to patients enrolled as suggested.
- suggest changing outcome to endpoint or even better primary endpoint if this is what is reported in the table
Authors: Changed to primary endpoint as suggested.
Reviewer 3 Report
Concerning the context of the review, it should be called “The difficulties (but not “potential”) of therapeutic vaccination in glioblastoma”, since unfortunately, there are still no special achievements to date.

Author Response
Concerning the context of the review, it should be called “The difficulties (but not “potential”) of therapeutic vaccination in glioblastoma”, since unfortunately, there are still no special achievements to date.
Authors: Thank you very much for reviewing our manuscript. We agree with this comment. Since no vaccine-based therapeutics are broadly available for use in glioblastoma with no major successes in the area, we revised our title to “The current status, challenges, and future potential of therapeutic vaccination in glioblastoma.”
Round 2
Reviewer 1 Report
no comments
Author Response
Thank you very much for reviewing updated version of our manuscript.